# Application of Starch Based Coatings as a Sustainable Solution to Preserve and Decipher the Charred Documents

Sonali Kesarwani [1] , Divya Bajpai Tripathy [1,]*  and Suneet Kumar [2]

1   School of Basic and Applied Sciences, Galgotias University, Greater Noida 201312, Uttar Pradesh, India
2   Forensic Science Laboratory, Moradabad 244001, Uttar Pradesh, India
*   Correspondence: divya.tripathy@galgotiasuniversity.edu.in

**Abstract:** Fire can be one of the most destructive elements to cause devastation. Fire can completely or partly destroy any crucial and invaluable documents, such as banknotes, books, affidavits, etc., in a couple of minutes. Moreover, the documents can also be damaged by heat, smoke, soot, and water during an accident. The burnt documents become fragile, losing their identity, which may have some evidentiary value related to the incident. Therefore, there is a strong need for processing to procure, preserve, and decipher, i.e., to restore the texts written on them. Hence, the present research focuses on developing a new method using natural polysaccharides, i.e., starch, to preserve and decipher the contents of charred documents. The most suitable concentration of starch analog was found to be 6% microwaved at 80 °C for about 10 min. As soon as the charred documents were coated with 6% starch analog, the majority of the invisible texts became visible to the naked eye in a second. Moreover, the application of a synthesized analog of polysaccharide on fragile charred documents provided an appreciable increase in strength by almost 0.1 kg/cm$^2$ for the coated charred documents of each paper type compared to that of non-coated ones and made them stabilized. This research also involves the use of easy and advanced handwriting recognition techniques (HCR) using an easily accessible, free platform, G-lens, that successfully recognized the majority of texts deciphered using 6% starch analog and converted them from captured images to a readable and copyable text format. Furthermore, the document visualization under VSC also gave a promising result by enhancing and deciphering the non-visible and less visible texts under flood light and white spot light at 715 and 695 long passes. Hence, this study offers an environmentally friendly, cost-effective, and sustainable approach of using a natural polysaccharide instead of synthetic polymers for the preservation and decipherment of charred documents.

**Keywords:** natural polysaccharide; questioned document; charred; decipherment; character recognition





## 1. Introduction

In arson cases or other fire incidents, documents, both printed and handwritten with writing instruments (generally pen), can be destroyed either accidentally or purposely due to excessive heat and smoke. During a fire incident, one or more paper placed together may burn either completely or partly due to a limited oxygen supply. The deliberately or accidentally burned documents may have an evidentiary value and may provide proof to link a crime with a criminal. Often, such documents may not completely turn into ashes due to several factors that may prevent the entire document from being burned; however, they may still make the document fragile, brittle, or black and render the writing invisible to the naked eye or unreadable, due to excessive smoke and soot from the fire [1]. Such documents need to be handled and preserved with utmost care, and the crucial content needs to be deciphered using a suitable method.

Decipherment of charred documents refers to analyzing and interpreting texts that have been damaged by fire or other heat sources. It is a specialized field of study that requires a combination of knowledge in linguistics, paleography, and chemistry. Charred

documents can be found in various contexts, such as archaeological sites, historical archives and crime scenes. They may contain valuable information about ancient cultures, historical events, or personal records. However, their decipherment is often challenging because the heat can cause the ink or writing material to evaporate or fuse with the surface, making the text illegible. To decipher charred documents, experts may use a variety of techniques, such as multispectral imaging, chemical analysis, and comparative analysis with known texts. They may also rely on their knowledge of the document's language, script, and historical context. The decipherment of charred documents is important because it can reveal new insights into the past that would otherwise be lost. It can shed light on historical events, social practices, and linguistic evolution. It can also provide a deeper understanding of the people who created the documents and the cultural context in which they lived.

Before the 20th century, writing inks contained traces of metals like iron and copper as tagging agents. Therefore, for deciphering the writing written with such inks, Blagden (1787) developed a method using potassium ferrocyanide to test the nature of the ink on ancient parchment for decipherment [2]. Later, Davis (1922) [3] proposed a method using a photographic plate to decipher the content of charred documents, while Mitchell (1925) used the calcining method, a process of further burning the carbonized fragment to decipher content written with pencil or some special inks, or content that was typewritten or printed [4]. Moreover, in 1935, Mitchell used infrared (IR) light with filters and plates to enhance charred documents' content [5]. Subsequently, Radley and Grant (1940) used fluorescent oil and ultraviolet light to successfully decipher the writing on printed matter, photocopies, typescripts, and carbon copies [6]. Similarly, many other methods have been developed, such as using chloral hydrate, a 5% solution of silver nitrate, an alcohol–glycerin solution, etc., to decipher the content of charred printed and typewritten documents. Over time, the development in ink composition led to iron being removed as a constituent because of its rusting and corrosive effect on the nib of pens, hence damaging the paper on which such pens were used. Therefore, current ink compositions exclude any traces of metals and only include pigments, dyes, resins, glycerol, alcohol, oils, and fats [7]. Thus, when documents written using such inks are charred, their components like pigments, dyes, alcohol, resins, etc., burn out, leaving lubricants like oils and fats on the surface of the charred documents because of their high boiling point. This aids in the decipherment of writing by different means. In questioned document examination, the major challenge faced by forensic practitioners is the handling and stabilization of highly fragile and brittle charred documents, hence rendering the decipherment of crucial information possible.

Deciphering charred documents using thin coatings is an established technique known as "palimpsest imaging". The term "palimpsest" refers to a manuscript or piece of writing on which the original writing has been erased or obscured, and new writing has been added on top. In order to decipher charred documents, a thin coating of some specific materials such as gold or silver is applied to the surface of the document. This coating reflects light differently depending on the depth of the writing and can expose previously obscured text. The process is based on the principles of reflection of light on the document at a certain angle, which creates shadows where the text is raised. These shadows can be captured using a high-resolution camera and can then be further processed using specialized software to enhance the contrast between the text and the background. This technique has been successfully used to decipher ancient manuscripts, including the Archimedes Palimpsest, a 10th-century manuscript containing the only surviving copies of some of Archimedes' works. The technique has also been used to reveal text on charred documents from the Villa of the Papyri in Herculaneum, which was destroyed because of the eruption of Mount Vesuvius in 79 AD. However, palimpsest imaging is a powerful tool for deciphering charred documents but the exploitation of costly metals makes this procedure cost-ineffective. Moreover, this technique is limited to document decipherment and is unsuitable for their preservation.

Other chemicals used for this purpose are polyvinyl acetate (PVA), methyl methacrylate (Bed acryl) [8], acetone, alkyl-2-cyanoacrylate ester (superglue fuming) [9], ammonia solutions [10], etc. Moreover, according to Harrison, the use of PVA may hinder decipher-

ment under infrared (IR) and ultraviolet (UV) light and also requires some preprocessing before applying PVA over charred documents [11,12]. Additional challenges associated with this approach are the chemical nature, toxic effect, low resistance to weather and moisture, poor resistance to most solvents, slow setting speed, creeping under substantial static load, etc. [11]. The application of PVA on charred documents is also a challenging task. Various researchers suggested the Pasteur pipette as a suitable and appropriate means of PVA application, while others recommended the use of a fine mister, such as a re-used perfume sprayer. In both the cases, significant caution and care need to be taken while applying the PVA in acetone solution to avoid further tempering to the charred documents due to their extremely fragile nature [13].

To overcome these challenges, researchers explored and developed an improved preservative and better procedure for its application over charred documents: natural polysaccharide coating as a green, non-toxic, cost-effective approach. This method also offers a fast and non-tedious synthetic and application technique to preserve the charred documents [14]. Moreover, transparent texture (for good visibility of texts), appropriate viscosity (for ease in application), good decipherment properties, short drying duration, and ability to provide appreciable strength to the coated charred documents were also the mandatory requisites that were found with starch-based coatings.

Polysaccharides, one of the most plentiful natural polymers, have the potential to take the place of "petroleum-based polymers", which are challenging to degrade in paper coatings. Polysaccharide molecules have a large number of hydroxyl (-OH) groups that can bind strongly with paper fibers through hydrogen bonds. Furthermore, their chemical modification can also effectively improve the mechanical barrier, thermal resistance, and hydrophobic properties of polysaccharide-based coatings and make them suitable to coat charred paper. In addition, polysaccharides can also give additional functional properties to the paper by dispersing and adhering functional fillers, e.g., conductive particles, catalytic particles, or anti-microbial chemicals, onto the coated paper surface. Starch is a polysaccharide made of two types of $\alpha$-D-glucan chains: amylose and amylopectin. Starch molecules produced by each plant species have specific structures and compositions (such as the length of glucose chains or the amylose/amylopectin ratio), and the protein and fat content of the storage organs may vary significantly. The majority of natural starches are amylopectin, which has film-forming ability, but its film mechanical properties still need to be improved [15]. When used as a paper coating, pure starch still has some other drawbacks. For instance, starch is sensitive to water vapor and usually forms a brittle coating layer [16]. Pure starch also forms faults in coating layers because of residual air, which results in large surface pores. To avoid all these problems, starch is usually modified through gelatinization and etherification [17–19]. Starch is a natural biopolymer that has been extensively used in various industries such as food, pharmaceuticals, and papermaking due to its abundance, low cost, and biodegradability [20]. Moreover, it is also used in the archaeological conservation of documents and artifacts to shield them from further deterioration [21]. Starch and its derivatives are the most frequently used sizing agents [22]. Sizing is used to improve the absorption and wear characteristics of paper. The most common materials used in sizing solutions are starch, latex, polyvinyl alcohol (PVA), and carboxymethyl cellulose (CMC). Surface sizing requires modification of native starch to achieve low viscosity of the starch solution. Many kinds of modified starches, such as acid-degraded starch, oxidized starch, starch ester, cationic starch, enzyme-degraded starch, and carboxymethyl starch, are used for surface sizing [23]. Starch ether is synthesized through the reaction of native starch with alcohol in an acidic medium using a process called etherification (Figure 1) [24–26].

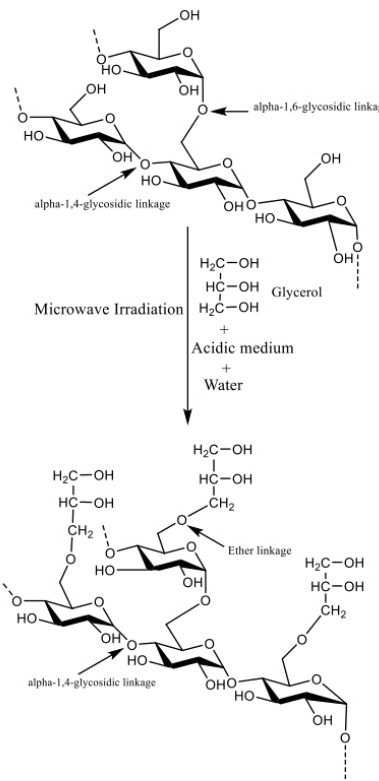

**Figure 1.** Scheme for the synthesis of partially etherified starch [26].

Starch-based films are thin, transparent films made from starch and other biopolymers, and are used in packaging materials, paper coatings, and other applications. Recently, starch-based thin films have gained increasing attention over traditional synthetic films due to their unique properties such as low cost, high transparency, biodegradability [20,25], renewability, and excellent mechanical properties. They have received great attention due to their sustainable and environmentally friendly characteristics [26] and have been proven as an alternative to petroleum-based and synthetic coatings, which have become a significant source of pollution and environmental degradation. Such benefits and unique properties of starch-based thin films make them suitable for various applications in different industries [27].

In the current work, we have explored a new and sustainable alternative to preserve and decipher charred documents. In this work, starch, a natural polymer, has been exploited to form a coating material. The process involves the creation of a hydrophobic thin film over the charred document and results in a suitable deciphering agent for the documents without even using any spectroscopic or analytical techniques. In order to adopt green synthetic pathways and a quick mode of synthesis, microwave-assisted synthesis procedures were adopted. The characterization of the synthesized starch analog (coating material) was achieved using attenuated total reflectance–Fourier transform infrared spectroscopy (ATR-FTIR) [28], and the effect of the physical properties of the synthesized analog on the mechanical strength of coated charred documents was tested with respect to non-coated charred documents. This work also validated the charred document decipherment using an easily available handwritten character recognition approach (HCR) [29] via Google Lens (G-lens). The present study holds great importance in the field of forensic document examination as well as historical document conservation. In forensic science, the application of starch-based coating for the preservation and decipherment of charred documents and for the retrieval of valuable content enables the use of these preserved and stabilized fragile charred documents as a piece of evidence in a court of law, related to any offense. On the other hand, document conservators can use this technology to preserve historical documents.

## 2. Materials and Methods

In this study, soluble laboratory starch (extra pure), manufactured by Sisco Research Laboratories Pvt. Ltd. (SRL, Delhi Depot, India), glycerol, acetic acid, and distilled water were used, and the research was conducted in the Department of Forensic Science, Galgotias University.

### 2.1. Sample Preparation

Charred Samples

Samples were made using a Linc Pentonic brand blue ballpoint pen on 3 different types of papers, viz., 75 g/m$^2$ A4 size white JK copier paper, 80 g/m$^2$ A4 size white JK copier paper, and bond paper used for affidavits (e-stamp). Each type of paper was cut into four equal parts (15 × 10.5 cm) and 10 samples of each paper type were prepared, generating a total of 30 samples. A predetermined paragraph was written by the same person on each sample, since different people apply different pressure, which may impact the results.

The prepared samples were charred in a muffle furnace (Thermotech, Faridabad 121001, Haryana) [10] at a temperature ranging from 280 to 310 °C and then removed and placed in a box for safekeeping. The documents charred at temperatures below 300 °C had written contents visible on them. Different grades of paper reach maximum charring with invisible texts at different temperatures (Table 1). Therefore, the samples were charred to the point when they became dark brown to black-grey with the writing completely invisible and were too fragile to handle. In the case of 75 g/m$^2$ copier paper, it reached maximum charring at 300 °C, 80 g/m$^2$ copier paper charred at 302 °C, and the bond paper at 305 °C, as shown in Figure 2 In each case, below this temperature, the sample was not appropriately charred, and the writing was visible, and above this temperature, the samples started igniting and gradually turning into ashes. Each sample had its picture taken using a mobile camera (OnePlus Nord CE 5G) for a before and after comparison.

**Table 1.** Effect of temperature on different types of paper.

| Paper Type | Temperature (°C) | Color and Effect on Paper | Effect on Written Text |
|---|---|---|---|
| 75 g/m$^2$ A4 size white JK copier | 280 | Light brown to white | Visible |
| | 290 | Brown in major areas while light brown in some places | Faintly visible in some places |
| | 300 | Dark brown to black with slightly curly edges | Completely invisible |
| | Above 300 | Black to grey with gradual ignition from edges, turned into ashes | Completely invisible |
| 80 g/m$^2$ A4 size white JK copier | 290 | Light brown to white in a half–half area | Visible |
| | 300 | Brown in the major area | Faintly visible in some places |
| | 301 | Completely brown in full area | Invisible |
| | 302 | Dark brown to black with curly edges | Completely invisible |
| | Above 302 | Black to grey with ignition from edges, turning into grey ashes | Completely invisible |
| A4 size bond paper Affidavit (e-stamp) | 300 | Light brown to white in half the area | Visible |
| | 302 | Brown in the major area | Faintly visible |
| | 304 | Dark brown in full area | Very faintly visible |
| | 305 | Dark brown to black | Completely invisible |
| | Above 305 | Black to grey, ignited, turned into ashes | Completely invisible |

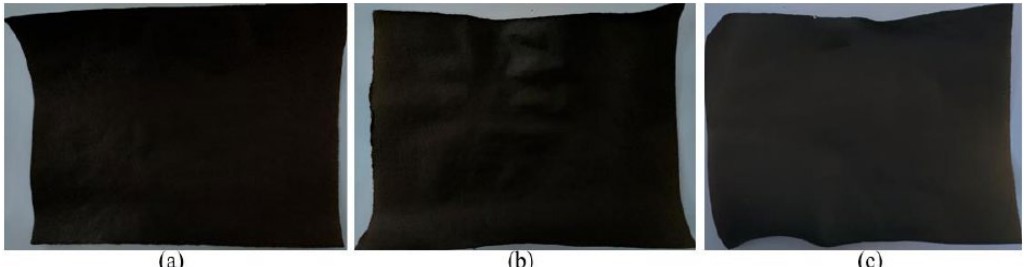

**Figure 2.** Charred samples: (**a**) 75 g/m$^2$ (300 °C), (**b**) 80 g/m$^2$ (302 °C), and (**c**) bond paper (305 °C) made with Linc Pentonic blue ballpoint pen showing invisible and unreadable texts.

### 2.2. Method of Preservative Preparation

Microwave Synthesis of Starch Analog

Varied-concentration starch solutions (2%, 4%, 6%, and 8%) [30] were made by mixing 2 g, 4 g, 6 g, and 8 g of starch in 100 mL distilled water, respectively, and adding 1 mL glycerol and 1 mL acetic acid to each [31] to optimize the appropriate concentration, as shown in Table 2. The relation between starch and glycerol is approximately in the ratio 1:3. The advantage of using a dilute solution of acetic acid is the low environmental impact compared with other acid solutions [31].

**Table 2.** Optimization of starch analog with the observed properties.

| Concentration (*w/v*) | Result |
| --- | --- |
| 2% | Transparent, low viscosity, not providing appreciable coating, strength, and decipherment |
| 4% | Transparent, viscosity not up to mark, not providing appreciable strength and decipherment |
| 6% | Transparent, appreciable viscosity was suitable for the application, providing appreciable strength and decipherment |
| 8% | Opaque and cloudy, high viscosity, white flaky appearance, not suitable for application and decipherment, cracks observed after drying |

Each starch solution was then subjected to microwave irradiation at a temperature of 80 °C and power of 462 watts for about 10 min with constant stirring in the middle until a clear, transparent, sticky analog of starch was acquired.

### 2.3. Spectral Characterization

The reactants, pure starch powder, glycerol, acetic acid, and the synthesized starch analog, were subjected to ATR-FTIR (Bruker) for spectral characterization to test the presence of different band stretching and the possible formation of the new analog peak after the reaction of starch and glycerol molecules in the acidic medium. The Bruker ATR-FTIR was equipped with a diamond ATR crystal. Prior to the sample analysis, a background spectrum was taken to account for any instrumental or environmental noise by setting the desired measurement conditions (wavenumber 4000–400 cm$^{-1}$) similar to the sample analysis. Thereafter, the samples were analyzed to acquire their spectral measurements. The spectrum was recorded on the basis of the percentage transmittance (%T) against the wavenumber (cm$^{-1}$). The measurement was repeated thrice to ensure the reproducibility of the result.

### 2.4. Application of Starch Analog

Each prepared starch analog was carefully applied over each sample paper type by pouring about 0.5 mL of starch analog over the charred document and spreading it with the help of a small handheld wiper. The wiper used in the application of the starch analog

was sufficiently flat, having a smooth silicon base to apply a smooth, evenly coated layer of preservative to the paper surface. The remaining charred samples of each paper type were kept aside for the purpose of comparing their physical properties and the visibility of texts before and after application in coated and non-coated samples.

### 2.5. Handwriting Character Recognition (HCR)

Soon after the application of the synthesized analog, images of the coated samples were captured using a OnePlus Nord CE 5G mobile camera. Thereafter, they were subjected to handwriting character recognition using Google Lens on the same mobile phone. The captured images were then searched and converted into readable and copiable texts.

### 2.6. Video Spectral Comparator (VSC)

The samples were visualized under an advanced optical instrument: a video spectral comparator, which incorporates various light sources, widely employed for document examination in forensic science laboratories. The samples were placed in the chamber of the VSC and then were visualized under VIS light, flood light, and white spot light at varied long passes to achieve the maximum visibility of texts.

### 2.7. Mechanical Properties
#### 2.7.1. Paper Folding Test

The mechanical properties of the coated and non-coated charred documents were preliminarily tested by estimating the increase in the strength of the coated charred documents compared with that of the fragile non-coated charred documents. After the coated charred samples had dried completely, they were subjected to a paper folding test to check the strength. Firstly, the non-coated charred samples were folded at the edges along the sides, and then the same technique was followed with samples coated with the synthesized analog.

#### 2.7.2. Bursting Strength Test

The confirmatory test for the increase in strength of the coated and non-coated charred documents was performed using a Pacorr digital bursting strength tester. This instrument gives quantitative data on the strength of a sample. The bursting strength was calculated based on the force applied by the plunger to the sample under examination, causing it to rupture. Similarly, each non-coated and charred document coated with 6% starch analog was tested to obtain a before-and-after comparison and measure the increased stability.

## 3. Results and Discussion

In an attempt to preserve and decipher charred documents, it is crucial to note the fact that different documents vary in their physical and chemical composition based on the variety of paper and ink used to make such documents, and also upon the conditions in which they became charred [32]. Therefore, the results of the present research were based on the thickness, i.e., gram per square meter ($g/m^2$), of the paper, the type of ink used to write the statement, and the coating material used to preserve and decipher the charred document.

As shown in Figure 2, the samples made using different paper types reached maximum charring at different temperatures. The 75 $g/m^2$ paper reached maximum charring at 300 °C, 80 $g/m^2$ at 302 °C, and bond paper at 305 °C. Once charred, the documents had invisible text and were too fragile and brittle enough to handle. Thus, they were stabilized and preserved using the synthesized starch analog.

The spectral characterization using ATR-FTIR of reactants (starch powder, glycerol, and acetic acid) and the synthesized analog are shown in Figure 3. Figure 3a shows the ATR-FTIR spectra of pure starch powder, which corresponds to α-1,4-glycosidic linkage at 1143.78 $cm^{-1}$ [33] and α-1,6-glycosidic linkage at 990.79 $cm^{-1}$ and 851.51 $cm^{-1}$ [34], which represent the skeletal model of the amylopectin starch ring and the major -C-OH

stretching of the primary and secondary alcoholic group at 3748.08 cm$^{-1}$ and 3271.32 cm$^{-1}$, respectively (Table 3).

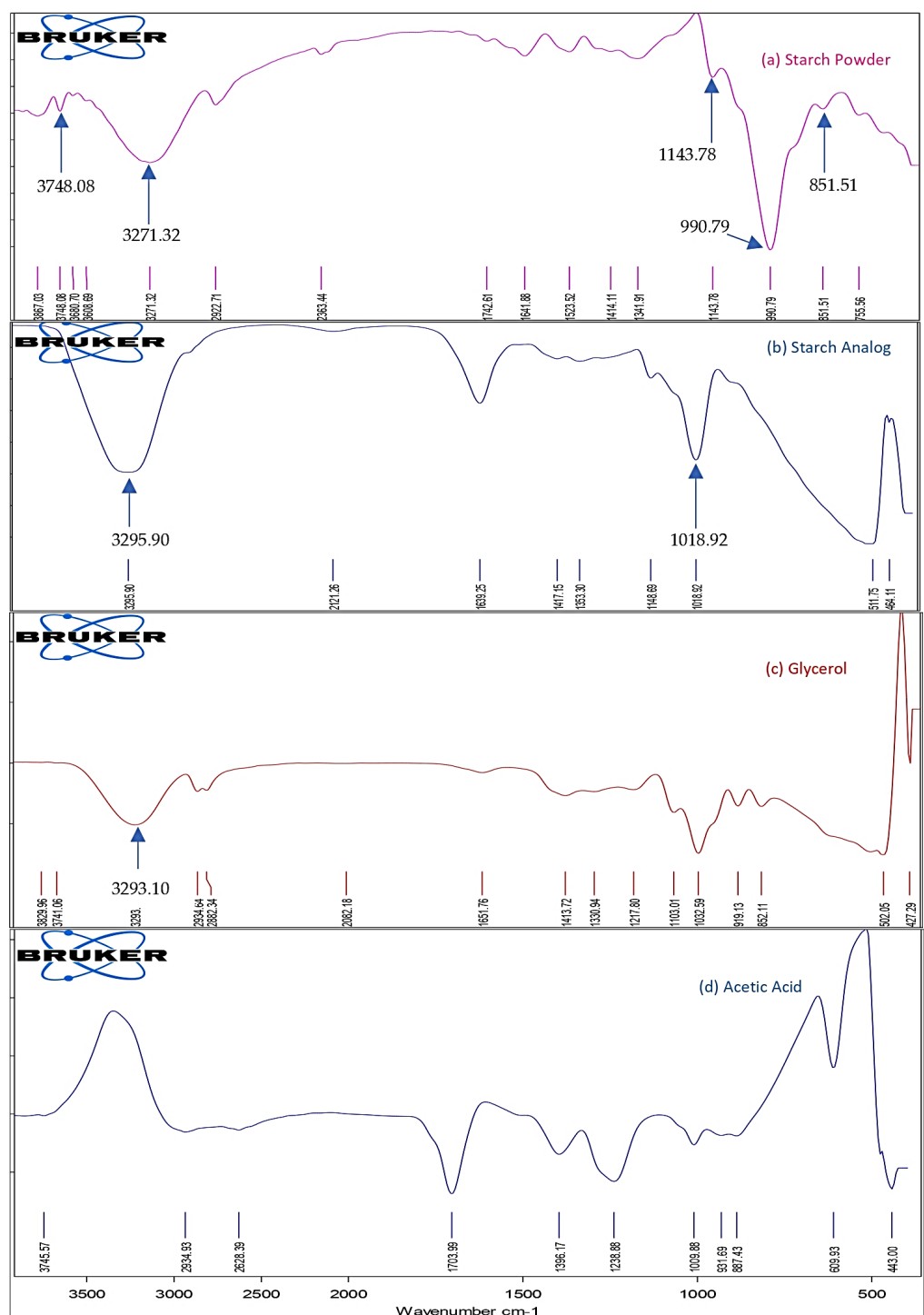

**Figure 3.** ATR-FTIR spectra of (**a**) starch powder, (**b**) synthesized starch analog, (**c**) glycerol, and (**d**) acetic acid.

The spectral peak at 1018.92 cm$^{-1}$ in Figure 3b indicates the formation of ether linkages (C-O-C) in the starch analog other than glycosidic bonds with the involvement of the C3 hydroxyl group of glycerol and one of the C6 carbons of the monosaccharide moieties of starch. After the reaction between starch and glycerol in the acidic medium, the hydroxyl (-OH) stretching in the starch analog shifted from 3271.32 cm$^{-1}$ to 3295.90 cm$^{-1}$, and a band

of glycosidic bonds shifted from 990.79 cm$^{-1}$ to 1018.92 cm$^{-1}$, which corresponds to C-O-C stretching and confirms the formation of etherified starch [35–37]. Hence, the result from the ATR-FTIR spectra of the synthesized starch analog shows that partial etherification occurred during a condensation reaction between starch and glycerol. The amount taken for starch and glycerol is in the ratio of 1:3, with glycerol being the limiting agent that carries out partial etherification of starch [38].

**Table 3.** Major spectral stretching with their wavenumber.

| Bond Stretching ↓ | Wavenumber (cm$^{-1}$) |
|---|---|
| **Starch Powder** | |
| $\alpha$-1,4-glycosidic | 1143.78 |
| $\alpha$-1,6-glycosidic | 990.79 & 851.51 |
| 1°-OH | 3748.08 |
| 2°-OH | 3271.32 |
| **Starch Analog** | |
| -OH | 3295.90 |
| C-O-C Stretch | 1081.92 |
| **Glycerol** | |
| -OH | 3293.10 |

In arson cases, documents charred at high temperatures are dehydrated and become brittle and weak. Moreover, the texts on them also vanish due to the evaporation of the volatile solvents of the ink composition, which makes the writings on documents invisible. However, the residue of inks components like fats and oils may still be present on the charred documents but invisible. The disappearance of written texts and the blackening of documents due to excessive heat and smoke become a serious problem for the document under decipherment [39]. On application of the optimized starch analog, it was found that in normal daylight, soon after the application, the texts became visible to the naked eye in most of the places of the charred document, which were previously, i.e., before preservative application, invisible to the naked eye as well as in oblique lighting. This may be due to the ballpoint pen ink composition and due to the removal of carbon and smoke particles embedded in the grooves made by the pen pressure on the charred samples.

Figure 4 shows the coated charred documents with 2% starch analog. After applying starch analog, it did not give appreciable results as the hydrophobic coating did not form on the samples and was not dry even after exposure to the air for hours. Once it dried, a thin film formed, which could not provide appreciable strength to the charred samples, as tested by the paper folding test. The decipherment of text was also not sufficient as none of the text could be read by the naked eye in the case of the bond paper coated with 2% analog, while only a few letters of the statement were read on the 75 g/m$^2$ and 80 g/m$^2$ paper. The same results were confirmed using HCR via Google Lens (G-lens) (Figure 5). This may be due to the lower viscosity of the analog, which could not provide enough coating and hence the decipherment of the invisible texts. Similarly, the application of 4% starch analog (Figure 6) also did not provide promising results, as only a few texts could be read in the case of the 80 g/m$^2$ and bond paper coated with 4% analog, while the majority of texts were deciphered on the 75 g/m$^2$ paper, which was confirmed using the HCR technique through G-lens (Figure 7).

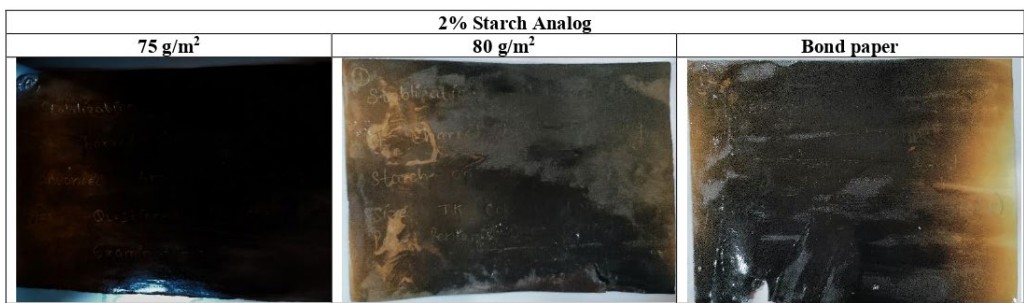

**Figure 4.** Coated charred samples with 2% starch analog on 75 g/m², 80 g/m², and bond paper (affidavit) written with Linc Pentonic blue ballpoint pen.

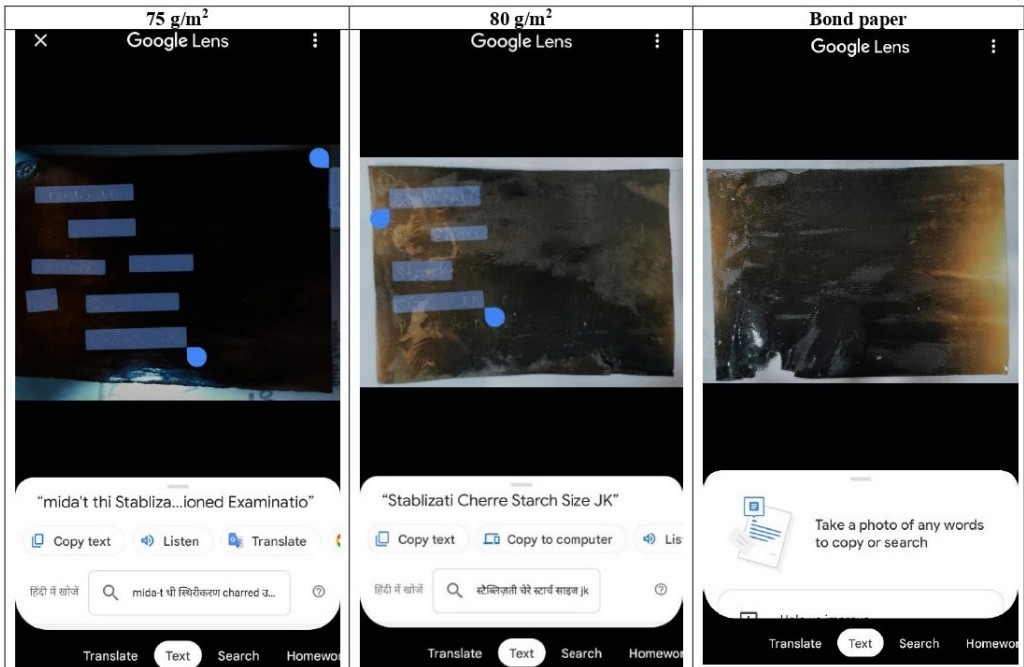

**Figure 5.** HCR-read texts of coated charred samples with 2% starch analog on 75 g/m², 80 g/m², and bond paper (affidavit).

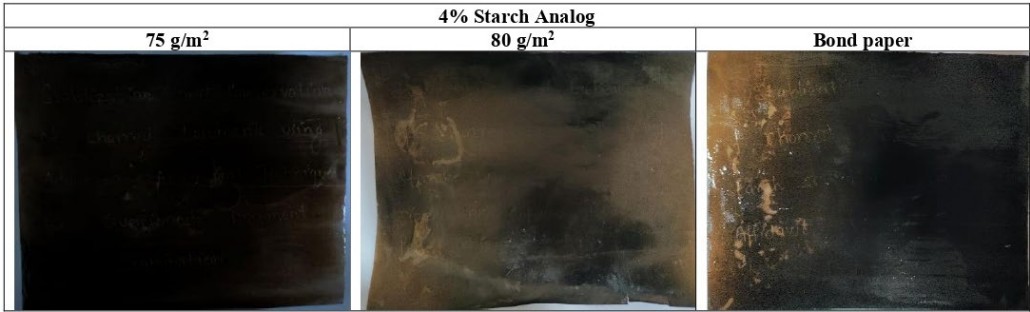

**Figure 6.** Coated and preserved charred samples with 4% starch analog on 75 g/m², 80 g/m², and bond paper (affidavit) written with Linc Pentonic blue ballpoint pen.

Moreover, 8% starch analog was found to be viscous enough to form an even coating on the charred document that resulted in the formation of white precipitated flakes of starch, which hindered smooth application. None of the text could be deciphered on the 75 g/m² and bond paper, while only some could be faintly seen on the 80 g/m² paper through the naked eye (Figure 8), as well as using G-lens (Figure 9). After it had dried, cracks of coating developed on the surface, and it curled.

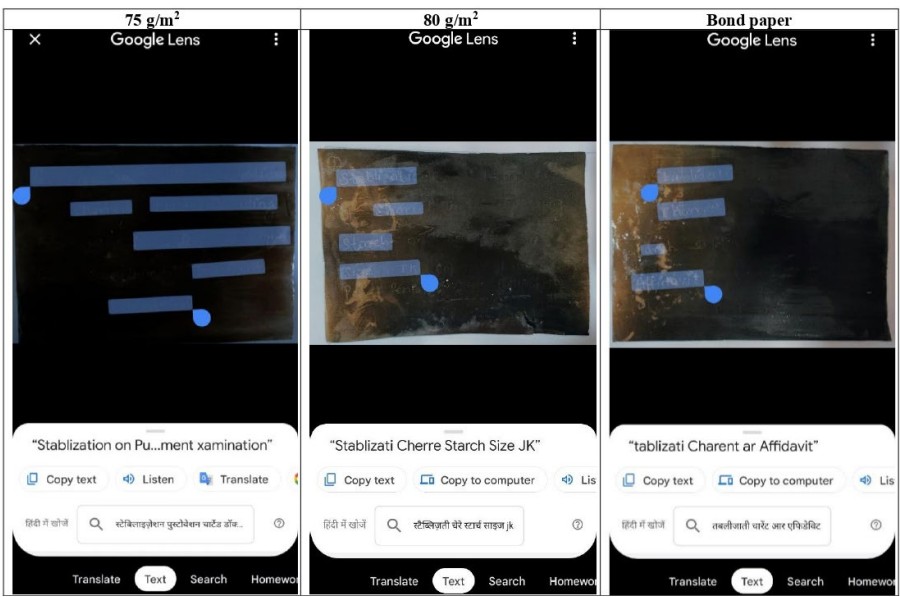

**Figure 7.** HCR-read texts of coated charred samples with 4% starch analog on 75 g/m², 80 g/m², and bond paper (affidavit).

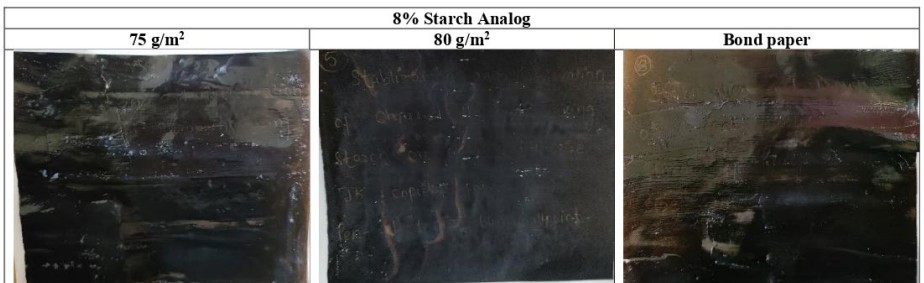

**Figure 8.** Coated and preserved charred samples with 8% starch analog on 75 g/m², 80 g/m², and bond paper (affidavit) written with Linc Pentonic blue ballpoint pen.

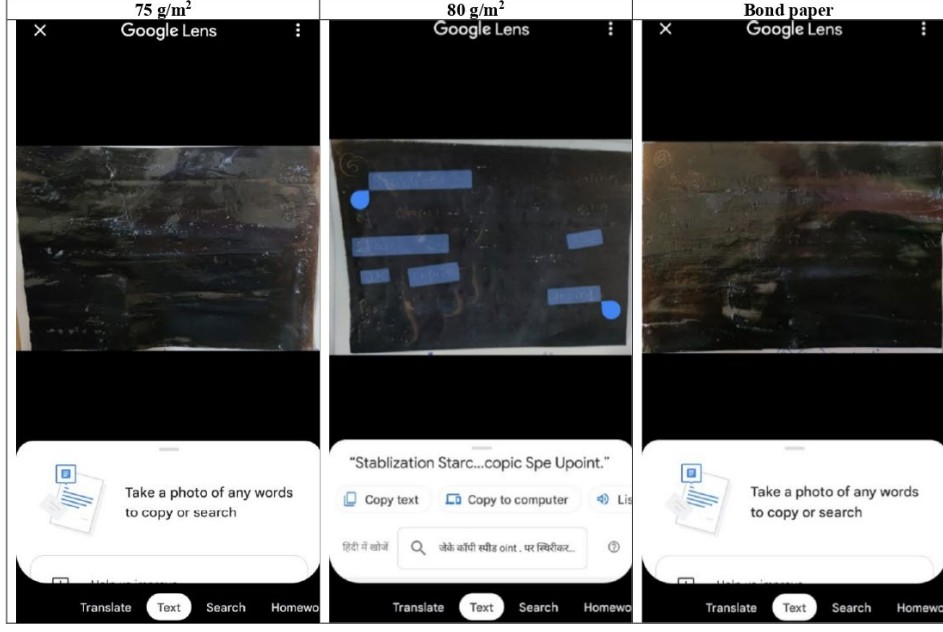

**Figure 9.** HCR-read texts of coated charred samples with 8% starch analog on 75 g/m², 80 g/m², and bond paper (affidavit).

In contrast, Figure 10 shows the application of 6% starch analog. The hydrophobic film formed like a polish that smoothed the surface and reduced the fragility and brittleness of the document through the process of rehydration. Soon after application, the texts were deciphered and visible to the naked eye. This may be due to the difference in polarity of the starch analog, consisting of glycerol, and ink composition, consisting of oils and fats. The residual traces of oils and fats on the burnt paper act as a repellent to the synthesized analog. Hence, the starch analog is absorbed by the charred paper except on the writing, leading to the differentiation due to color contrast between the writing and the background [10,40]. The HCR technique using G-lens also recognized the maximum deciphered text of the statement on all three types of paper, i.e., 75 g/m$^2$, 80 g/m$^2$, and bond paper (Figure 11). The coated charred document also dried in about 10–15 min at room temperature, which gave an extra advantage of this synthesized starch analog.

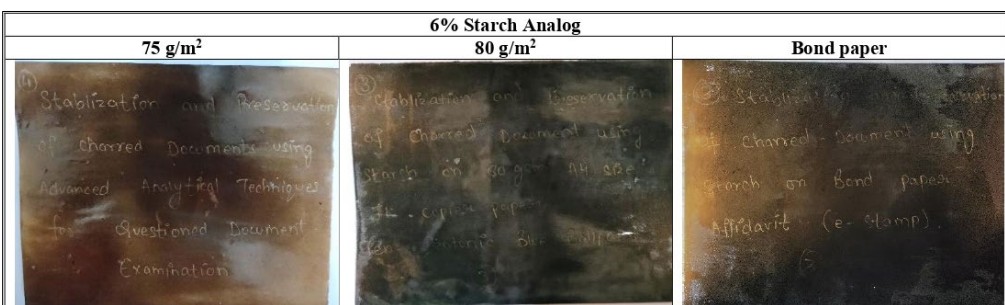

**Figure 10.** Coated and preserved charred samples with 6% starch analog on 75 g/m$^2$, 80 g/m$^2$, and bond paper (affidavit) written with Linc Pentonic blue ballpoint pen.

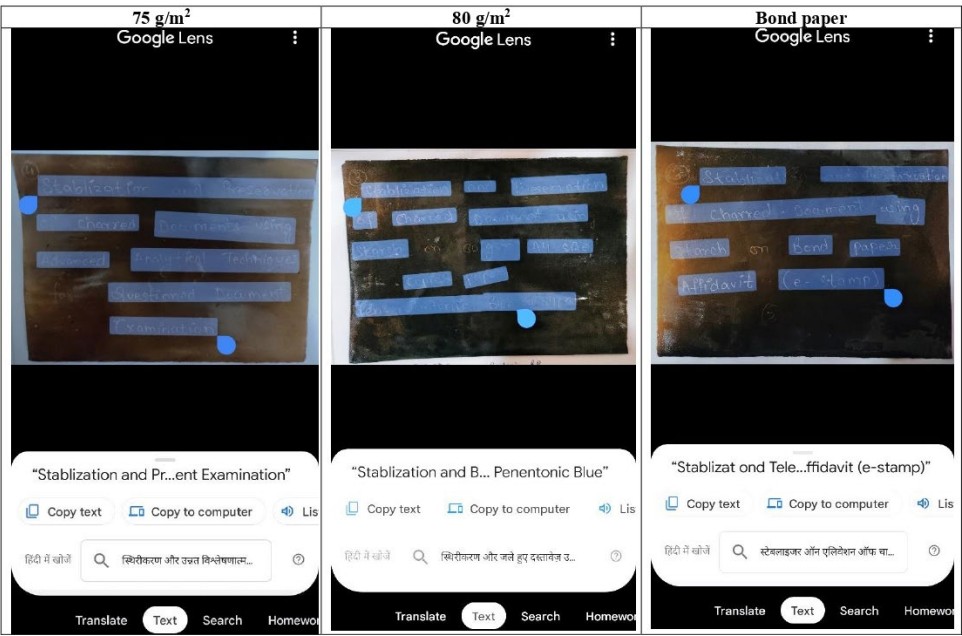

**Figure 11.** HCR-read texts of coated charred samples with 6% starch analog on 75 g/m$^2$, 80 g/m$^2$, and bond paper (affidavit).

The further analysis of the coated charred documents under an advanced optical VSC instrument with different light sources and at varied long passes gave quite good results of decipherment even a month after coating the charred document using starch analog. In the case of the 75 g/m$^2$ and 80 g/m$^2$ charred documents written with blue ballpoint pen ink, the texts were enhanced and deciphered under flood light at 715 long pass, whereas on the bond paper, the texts were more prominently legible under the white spot light at 695 long pass, as shown in Figure 12. This is due to the reflection of the different wavelengths

of light from the document surface under visualization, which makes the invisible and low-visibility text prominently visible.

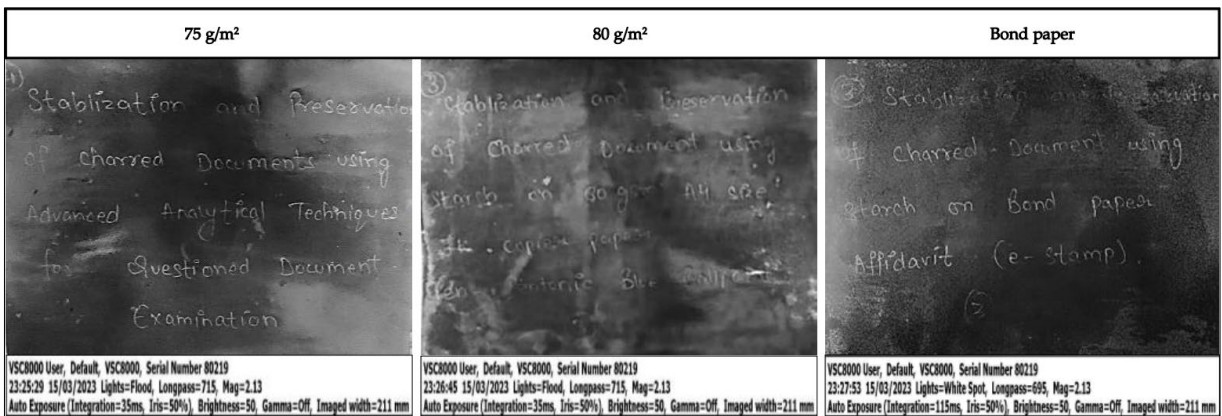

**Figure 12.** VSC-read texts of coated charred samples with 6% starch analog on 75 g/m$^2$, 80 g/m$^2$, and bond paper (affidavit).

Moreover, the mechanical properties of the coated charred documents showed that the coatings formed on the charred documents by applying the starch analog increased the strength of the fragile charred documents, which was tested preliminary by the paper folding test. Further, an experimental, qualitative analysis using a digital bursting strength tester calculated the bursting strength of the coated and non-coated charred documents, which provided promising results. The bursting strength of the 75 g/m$^2$ non-coated charred documents before stabilization was found to be 0.12 kg/cm$^2$, which increased to 0.21 kg/m$^2$ after stabilization with starch analog, whereas the bursting strength of the non-coated 80 g/m$^2$ and bond paper was 0.15 kg/cm$^2$ and 0.25 kg/cm$^2$, which increased to 0.25 kg/cm$^2$ and 0.35 kg/cm$^2$ after being stabilized with starch analog, as shown in Figure 13 [41,42]. This may be due to the penetration and absorbance of the starch analog on the charred layer of paper, which adhered to the surface, thus strengthening the fragile and brittle charred documents, which can be preserved and placed in safekeeping for a few more days.

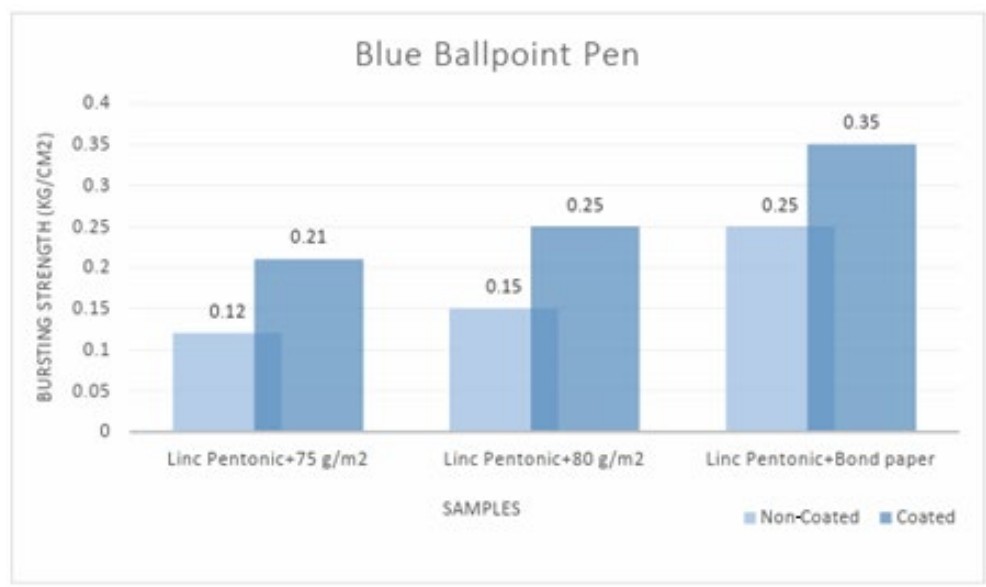

**Figure 13.** Graphical representation of the increase in bursting strength between non-coated and coated charred documents made using Linc blue ballpoint pens on 75 g/m$^2$, 80 g/m$^2$, and bond paper.

## 4. Conclusions

The present research concludes that the preservation and decipherment of texts of charred documents by any means depends on the type of paper, writing ink, condition, and level of charring. Soon after the application of 6% starch analog, the texts were visible on all three types of paper samples charred at varied temperatures. This gave the best result, since the texts could also be clearly recognized through handwriting character recognition (HCR) via Google Lens to check the readability level by converting the images to readable and copiable text. Moreover, it was found that the drying and setting (time) of the synthesized analog after application on charred samples took 10–15 min only. In addition, a protective layer of polysaccharide provided an increase in mechanical strength to the charred samples, which was observed and tested by general observation of thickness and the folding test method comparing coated and non-coated charred samples. Moreover, the sample dried, turned into a hydrophobic film, and was not affected by water, thus making the document moisture-protected. Hence, it can be concluded that for the documents made with a ballpoint pen preserved with 6% starch analog if charred up to around 300 °C (75 g/m$^2$), 302 °C (80 g/m$^2$), and 305 °C (bond paper affidavit), the writing can be deciphered and read easily, and the method is easy, non-toxic, and cost-effective. The present study suggests exploring the results based on more combinations of pens like gel pens, fountain pens, markers, etc., on different grades and colors of paper that are usually used in crucial documents. By delving into more combinations of writing instruments such as gel pens, fountain pens, etc., studies can expand their scope and investigate their impact on the preservation and decipherment of charred documents. This broader approach will provide a comprehensive understanding of how various writing tools interact with different grades and colors of paper during the charring process, particularly in the context of crucial documents. The outcomes of this extended research could potentially lead to the development of enhanced techniques and strategies for recovering and interpreting vital information from charred documents, thereby advancing the field of document analysis and historical research. Furthermore, these findings may have practical applications in forensic investigations and archival preservation, ultimately contributing to the preservation and understanding of invaluable historical records.

**Author Contributions:** Conceptualization, S.K. (Sonali Kesarwani) and D.B.T.; methodology S.K. (Sonali Kesarwani); software, S.K. (Sonali Kesarwani). and D.B.T.; validation, S.K. (Sonali Kesarwani) and D.B.T.; formal analysis, S.K. (Sonali Kesarwani).; investigation, S.K. (Sonali Kesarwani). and D.B.T.; resources, S.K. (Suneet Kumar).; data curation, S.K. (Sonali Kesarwani); writing—original draft preparation, S.K. (Sonali Kesarwani) and D.B.T.; writing—review and editing, S.K. (Sonali Kesarwani) and D.B.T.; visualization, S.K. (Sonali Kesarwani); supervision, D.B.T. and S.K. (Suneet Kumar); project administration, D.B.T. All authors have read and agreed to the published version of the manuscript.

**Funding:** This research received no external funding.

**Institutional Review Board Statement:** Not applicable.

**Informed Consent Statement:** Not applicable.

**Data Availability Statement:** No additional data are available.

**Acknowledgments:** The authors acknowledge the contribution of Anurag Singh, HBTU, Kanpur, and Vivek Kumar Bajpai, Department of Energy and Material Science Engineering, Dongguk University, Republic of Korea, for his additional guidance.

**Conflicts of Interest:** The authors declare no conflict of interest.

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
