# Peer review of "Application of Starch Based Coatings as a Sustainable Solution to Preserve and Decipher the Charred Documents"

_coatings, doi:10.3390/coatings13091521_

Round 1
Reviewer 1 Report
Reviewer comments
Kesarwani et al reported the application of starch as a sustainable coating material for charred documents preservation and decipherment. The present work is interesting to readers. The manuscript may be published after major revision.
1. The abstract part is very general. I suggest the author to change the abstract based on the obtained results.
2. The symbols of °C should be changed instead of 0C throughout the manuscript.
3. What is the importance of this work?
4. Why the authors choose only temperature and not other factors like time?
5. I suggest the author to provide the overall table for spectral characterization results
6. Is it possible for taking mass spectra for these samples
7. If it’s possible, I suggest the author to provide the results for blank (without starch).
8. I suggest the author to add future prospects of the work
9. Author should change the references based on the Journal format.
10. There is lack of discussion in this manuscript. I suggest the author to add more discussion with suitable references.
11. I found some typo error in this manuscript. I suggest the author to check the typo error throughout the manuscript.
Minor English language correction is required
Author Response
Response to Reviewer’s Comment 1
- The abstract part is very general. I suggest the author to change the abstract based on the obtained results.
Response: The abstract has been modified as per suggestion. L16-L29.
The most suitable concentration of starch analog was found to be 6% microwaved at 80 °C for about 10 min. As soon as the charred documents were coated with 6% starch analog, the majority of the invisible texts became visible to the naked eye in a second. Moreover, the application of synthesized analog of polysaccharide on fragile charred documents provided an appreciable increase in strength by almost 0.1 kg/cm2 to the coated charred documents of each paper type with that of non-coated ones that made them stabilized. The research also involves the use of easy and advanced handwriting recognition techniques (HCR) using an easily accessible, free platform G-lens, that successfully recognized the majority of texts deciphered using 6% starch analog and converted them from captured images to readable and copiable text format. Furthermore, the documents visualization under VSC also gave a promising result by enhancing and deciphering the non-visible and less visible texts under flood light and white spot light at 715 and 695 longpass. Hence, the study gives an environment-friendly, cost-effective, and sustainable approach of using a natural polysaccharide instead of synthetic polymers for the preservation and decipherment of charred documents.
- The symbols of °C should be changed instead of 0C throughout the manuscript.
Response: The symbol is corrected throughout the manuscript as suggested.
- What is the importance of this work?
Response: The present study holds great importance in the field of Forensic document examination as well as historical document conservation. In forensic science, the application of starch analog, which is a natural polysaccharide, for the preservation and decipherment of charred documents and retrieval of valuable content enables to use these preserved and stabilized fragile charred documents as a piece of evidence in a court of law that is related to any offense. On the other hand, document conservators can use this piece of art to preserve historical documents. L146-151.
- Why the authors choose only temperature and not other factors like time?
Response: The author choose only temperature during the charring process since the aim was also to study the effect of temperature on the charring of paper samples [10]. When studying the effect of temperature on the charring process, isolating one variable (temperature) allows the authors to directly observe and understand its influence without interference from other factors. This helps establish a clear cause-and-effect relationship. Therefore, the charring was done in a muffle furnace to study the effect of temperature in a controlled manner, that has only temperature controlling setting.
- I suggest the author to provide the overall table for spectral characterization results
Response: As per the suggestion, table 3 has been added to the revised manuscript. Moreover, the discussion on spectral characterization has also been modified.
- Is it possible for taking mass spectra for these samples
Response: Dear Sir, this time it is difficult to get the MASS SPECTRA of the sample as we don't have adequate time to get it done by an outside lab, and for this research, it is not very necessary as the emphasis is on the preservation and decipherment of the charred documents. But I will try to add the mass spectra in the supplementary file before the publication (if accepted).
- If it’s possible, I suggest the author to provide the results for blank (without starch).
Response: Thank you for your suggestion sir, I will try to add this in the supplementary file before the publication (if accepted).
- I suggest the author to add future prospects of the work.
Response: As per suggestion, the future prospects are added in the conclusion section L359-L369.
The present study suggests exploring the results based on more combinations of pens like gel pens, fountain pens, markers, etc., on different grades and colours of paper that are usually used in crucial documents. By delving into more combinations of writing instruments such as gel pens, fountain pens, etc., the study can expand its scope and investigate their impact on the preservation and decipherment of charred documents. This broader approach will provide a comprehensive understanding of how various writing tools interact with different grades and colours of paper during the charring process, particularly in the context of crucial documents. The outcomes of this extended research could potentially lead to the development of enhanced techniques and strategies for recovering and interpreting vital information from charred documents, thereby advancing the field of document analysis and historical research. Furthermore, these findings may have practical applications in forensic investigations and archival preservation, ultimately contributing to the preservation and understanding of invaluable historical records.
- Author should change the references based on the Journal format.
Response: The reference has been corrected as per the Journal format.
- There is lack of discussion in this manuscript. I suggest the author to add more discussion with suitable references.
Response: As per the suggestions, the discussion has been modified with more references.
L247-L252
As shown in Figure 3, the samples made using different paper types got maximum charred at different temperatures. The 75 g/m2 paper got maximum charred at 300 °C, 80 g/m2 at 302 °C, whereas bond paper at 305 °C. The documents once charred had invisible texts with fragile and brittle enough to handle. Thus, were stabilized and preserved using synthesized starch analog.
The spectral characterization using ATR-FTIR of reactants (starch powder, glycerol, and acetic acid) and the synthesized analog are shown in Figure 5.
L262-L267
The spectral peak at 1018.92 cm-1 in Figure 5b indicates the formation of ether linkages (C-O-C) in starch analog other than glycosidic bonds with the involvement of the C3 hydroxyl group of glycerol and one of the C6 carbon of monosaccharide moieties of starch. After the reaction between starch and glycerol in an acidic medium, the hydroxyl (-OH) stretching in starch analog shifted from 3271.32 cm-1 to 3295.90 cm-1 and a band of glycosidic bonds, shifted from 990.79 cm-1 to 1018.92 cm-1, which corresponds to C-O-C stretching, and confirms the formation of etherified starch [35-37].
L270-L275
Moreover, the texts on them also vanish due to the evaporation of volatile solvents of ink composition which makes the writings on documents invisible. However, the residue of inks components like fats and oils may still be present on the charred documents but invisible. The disappearance of written texts and the blackening of documents due to excessive heat and smoke become a serious problem with the document under decipherment [39]. On application of the optimized starch analog, it was found that in normal daylight, soon after the application the texts became visible at most of the places of the charred document with the naked eye, which was earlier i.e., before preservative application invisible to the naked eye as well as in oblique lighting (Figure 3).
L217-L226
The HCR technique using G-lens also recognized the maximum deciphered text of the statement in all three types of paper i.e., 75 g/m2, 80 g/m2, and bond paper (Figure 13). The coated charred document also dried in about 10-15 mins at room temperature which gave an extra advantage of the synthesized starch analog.
The further analysis of coated charred documents under an advanced optical instrument VSC at different light sources and at varied longpasses gave quite good results of decipherment even after a month of coating the charred document using starch analog. In the case of 75 g/m2 and 80 g/m2 charred documents written with blue ballpoint pen ink, the texts were enhanced and deciphered under flood light at 715 longpass, whereas in the bond paper, the texts were more prominently legible under the white spot light at 695 longpass, as shown in Figure 14. This is due to the reflection of the different wavelengths of light from the document surface under visualization that makes the invisible and low visibility of the text prominently visible.
L335-L342
Moreover, the coating of the starch analog provided appreciable strength to the charred document, tested preliminary by the paper folding test. Further confirmation through a digital bursting strength tester by calculating the busting strength of coated and non-coated charred documents provided promising results. The bursting strength of 75 g/m2 non-coated charred documents before stabilization was found to be 0.12 kg/cm2 which after stabilization with starch analog increased to 0.21 kg/m2, whereas the bursting strength of non-coated 80 g/m2 and bond paper was 0.15 kg/cm2 and 0.25 kg/cm2, which increased to 0.25 kg/cm2 and 0.35 kg/cm2 after getting stabilized with starch analog as shown in Figure 15 [41, 42]. This may be due to the penetration and absorbance of starch analog on the charred layer of paper which adhered to the surface, thus strengthening the fragile and brittle charred documents, that can be preserved and placed for little longer days in safekeeping.
- I found some typo error in this manuscript. I suggest the author to check the typo error throughout the manuscript.
Response: The corrections have been incorporated as per suggestion.
Reviewer 2 Report
The present study describes the Application of Starch as a Sustainable Coating Material for Charred Documents Preservation and Decipherment. Some of the results are relevant; however, the authors need to attend to the following observations:
- Revise and improve English grammar.
- The authors use FT-IR to investigate film composition. It is necessary to measure other properties of the film (uniformity and thickness by using AFM). Also, measure the mechanical properties.
- Please use a program (i.e., origin) for data analysis and graphing Figure 5.
- Film deposition must be reproducible (i.e., use the Dr. Blade technique). Report the amount of material used for the deposit.
Improve English grammar
Author Response
Reviewer 2
The present study describes the Application of Starch as a Sustainable Coating Material for Charred Documents Preservation and Decipherment. Some of the results are relevant; however, the authors need to attend to the following observations:
- Revise and improve English grammar.
Response: Correction has been made throughout.
- The authors use FT-IR to investigate film composition. It is necessary to measure other properties of the film (uniformity and thickness by using AFM). Also, measure the mechanical properties.
Response: The author used ATR-FTIR for spectral characterization of the raw materials used in the synthesis of starch analog, and to check the possibility of ether-linkage in a synthesized starch analog.
Moreover, the mechanical property like; the strength of the coated and non-coated charred documents was preliminary tested by general observation of their increase in thickness and paper folding test along the edges. Later on, the quantitative analysis of strength was also done using a digital bursting strength tester. The result demonstrated significant stability in terms of the strength of coated charred documents with that of fragile, brittle non-coated charred documents. However, it is not possible to check the mechanical strength of the film independently as the coating has been achieved by applying the liquid-melted analog on the charred document.
In the present work, the property of coated charred documents was evaluated only in terms of strength, since the emphasis of the study was to stabilize and preserve the fragile charred documents, and to decipher the invisible content of charred documents. Therefore, the other properties of the individual film were not a requisite field of the present study. However, as per the suggestion, the author will try to provide the data as a supplementary file.
- Please use a program (i.e., origin) for data analysis and graphing Figure 5.
Response: The spectra were plotted using Bruker ATR-FTIR software itself, and the result was generated in an image format from the outside lab. However, the marking of the spectral peaks has now been done in Figure 5.
- Film deposition must be reproducible (i.e., use the Dr. Blade technique). Report the amount of material used for the deposit
Response: As in the current research, charred documents were used which are already very fragile and brittle, hence the Dr. Blade technique was not found suitable. Moreover, each charred document was coated by pouring about 0.5 ml of synthesized starch analog and evenly spreading using a flat silicon-based wiper to give it a smooth coated layer of preservative and then was allowed to dry at room temperature. L257-L258.
Reviewer 3 Report
interesting text from a scientific point of view. Nevertheless, efforts should be made to refine it.
- photos no. 4 should be removed - it does not contribute anything to the study.
-figure 2 - should be removed, it also does not add anything to the study
- source of drawing no. 1 (own elaboration?)
- consider the IR spectra, Figure 5. Spectra c and d - measurements in transmittance or absorbance? Even in the case of absorbance, the spectrum cannot have negative values - whether the measurement was carried out on a calibrated apparatus or on an uncleaned ATR diamond.
- figure 10 - correct
- no description of the performance of IR analyses.
after corrections, the text should be re-evaluated
Author Response
Reviewer 3
Dear Sir,
Thanks for sparing your valuable time on my manuscript. Your valuable comments have definitely improved my research work presentation.
Please find below the pointwise responses:
Interesting text from a scientific point of view. Nevertheless, efforts should be made to refine it.
- photos no. 4 should be removed - it does not contribute anything to the study.
Response: Removed as per suggestion.
-figure 2 - should be removed, it also does not add anything to the study
Response: Removed as per suggestion.
- source of drawing no. 1 (own elaboration?)
Response: The source reference has been added in Figure 1 [26].
- consider the IR spectra, Figure 5. Spectra c and d - measurements in transmittance or absorbance? Even in the case of absorbance, the spectrum cannot have negative values - whether the measurement was carried out on a calibrated apparatus or on an uncleaned ATR diamond.
Response: L211-215
The Bruker ATR-FTIR was equipped with diamond ATR crystal. Prior to the sample analysis, a background spectrum was taken to account for any instrumental or environmental noise, by setting the desired measurement conditions (wavenumber 4000-400 cm-1) similar to the sample analysis. The ATR diamond crystal cleaned each time with ethanol after taking measurements of each reactant. Thereafter, the samples were analyzed to acquire their spectral measurements. The spectrum was recorded on the basis of the percentage transmittance (%T) against the wavenumber (cm-1). The measurement was repeated thrice to ensure the reproducibility of the result. None of the values showing is negative in this case.
- figure 10 – correct
Response: Corrected as per suggestion.
- no description of the performance of IR analyses.
Response: As per suggestion, the description of IR analysis is modified in the manuscript.
after corrections, the text should be re-evaluated
Response: The manuscript is reviewed thoroughly.
Reviewer 4 Report
The authors report on coating charred paper with starch and starch analog for their conservation and decipherment.
Using starch for coating paper and conserving charred paper, with the addition of thin rice paper, is a common practice in conservation and forensic fields. The authors should better present the novelty of their work since adding glycol-functionalities to starch is not innovative and does not add new knowledge to the field.
The authors report IR spectra of the chemicals and of the starch analog. The coated paper is not analyzed with any spectroscopic technique, only visual evaluation by consumer technologies is reported. The quality of the coating is not assessed with any analytical technique.
Fig.10 is split between pages 10 and 11, the authors should review the pdf version of their file. The same is valid for Fig.12.
The authors wish to assess forensic applications, but only google lens is used. No other camera, spectroscopy, or even different lightening is used to characterize the coated charred samples.
I suggest reconsidering the article publication after a thorough review of the text and further experiments to strengthen the authors' work. The novelty of the work should also be proved.
The quality of work and presentation are below standards. The novelty of the work is also not clearly presented.
Author Response
Dear Sir,
Thanks for sparing your valuable time on my manuscript. Your valuable comments have definitely improved my research work presentation.
Please find below the pointwise responses:
Reviewer 4
The authors report on coating charred paper with starch and starch analog for their conservation and decipherment.
- Using starch for coating paper and conserving charred paper, with the addition of thin rice paper, is a common practice in conservation and forensic fields. The authors should better present the novelty of their work since adding glycol-functionalities to starch is not innovative and does not add new knowledge to the field.
Response: In forensic science, particularly for questioned documents and charred document examination, the application of natural polysaccharides e.g., starch is a novel substitute for the synthetic polymer e.g., polyvinyl acetate (PVA) conventionally used for the preservation of charred documents. Moreover, the PVA only aid in the stabilization and preservation of charred documents, but does not give promising decipherment, rather PVA hinders decipherment under IR and UV light [11]. In addition, PVA being non-environment friendly, chemical nature, and non-biodegradable is an unsafe preservation technique. Therefore, the novel application of modified starch or starch analog using glycerol in a slightly acidic medium is novel in the field of charred document examination (Forensic science).
L146-151
Taking about the novelty of the present research, the study holds great importance in the field of Forensic document examination as well as historical document conservation. In forensic science, the application of starch analog, which is a natural polysaccharide, for the preservation and decipherment of charred documents and retrieval of valuable content enables to use these preserved and stabilized fragile charred documents as a piece of evidence in a court of law that is related to any offense. On the other hand, document conservators can use this piece of art to preserve historical documents.
- The authors report IR spectra of the chemicals and of the starch analog. The coated paper is not analyzed with any spectroscopic technique, only visual evaluation by consumer technologies is reported. The quality of the coating is not assessed with any analytical technique.
Response: The author used ATR-FTIR for spectral characterization of the raw materials used in the synthesis of starch analog, and to check the possibility of ether-linkage in a synthesized starch analog.
In the present work, the property of coated charred documents was evaluated in terms of strength, since the emphasis of the study was to stabilize and preserve the fragile charred documents, and to decipher the invisible content of charred documents. Therefore, the other properties of the individual film were not a requisite field of the present study.
Moreover, the mechanical property like; the strength of the coated and non-coated charred documents was preliminary tested by general observation of their increase in thickness and paper folding test along the edges. Later on, the quantitative analysis of strength was also done using a digital bursting strength tester. The result demonstrated significant stability in terms of the strength of coated charred documents with that of fragile, brittle non-coated charred documents. However, it is not possible to check the mechanical strength of the film independently as the coating has been achieved by applying the liquid-melted analog on the charred document.
- 10 is split between pages 10 and 11, the authors should review the pdf version of their file. The same is valid for Fig.12.
Response: The figures are now modified in jpg format.
- The authors wish to assess forensic applications, but only google lens is used. No other camera, spectroscopy, or even different lightening is used to characterize the coated charred samples.
Response: Soon after the application of synthesized starch analog, the invisible texts in the charred documents deciphered and became visible to the naked eye. The coated documents with deciphered texts were further undergone handwritten character recognition (HCR) technique using a Google platform, Google-lens. The analysis through G-lens gave appreciable results by recognizing the majority of deciphered words and characters and converting the captured image into a copiable text format. The G-lens software was used in the primary step due to its merits like being easily accessible anywhere, free service by Google, and easy to process without using any sophisticated instrument.
Furthermore, the coated charred documents were analyzed using an advanced optical instrument, Video Spectral Comparator (VSC) which incorporates various light sources, widely employed for document examination in Forensic science laboratories. The examination from VSC also gave promising results by deciphering and enhancing the majority of words and characters of the statement. The results are shown in the manuscript.
L320-L326
The further analysis of coated charred documents under an advanced optical instrument VSC at different light sources and at varied longpasses gave quite good results of decipherment even after a month of coating the charred document using starch analog. In the case of 75 g/m2 and 80 g/m2 charred documents written with blue ballpoint pen ink, the texts were enhanced and deciphered under flood light at 715 longpass, whereas in the bond paper, the texts were more prominently legible under the white spot light at 695 longpass, as shown in Figure 14. This is due to the reflection of the different wavelengths of light from the document surface under visualization that makes the invisible and low visibility of the text prominently visible.
- I suggest reconsidering the article publication after a thorough review of the text and further experiments to strengthen the authors' work. The novelty of the work should also be proved.
Response: Respected Sir, as per the suggestion, the authors have added more experimental results of the study, including the novelty of the work.
Round 2
Reviewer 1 Report
The authors rectified all the comments as per the reviewer suggestion. And so this manuscript can be published in Coatings.
Minor editing of English language like typo errors should be rectified.
Author Response
Reviewer 1
Minor editing of English language like typo errors should be rectified.
Response: Rectified as per suggestion.
Reviewer 2 Report
The authors did not attend to reviewers' comments. It was asked to improve the quality of Figure 5. Also, it is necessary to report some of the mechanical properties of the films. The authors indicate the inclusion of supplementary information, but no supplementary file exists. The authors must investigate the quality of the films and their reproducibility.
Minor revision
Author Response
Reviewer 2
The authors did not attend to reviewers' comments.
Response: We have tried to address each comment with due consideration.
It was asked to improve the quality of Figure 5.
Response: As per the suggestion, the quality of the Figure has been improved.
Also, it is necessary to report some of the mechanical properties of the films.
Response: As per the comments of the respected reviewer, the mechanical property of coated charred documents using starch analog has been added to the manuscript. L286-L301 and L426-439.
The authors indicate the inclusion of supplementary information, but no supplementary file exists.
Response: The supplementary information has been added to the manuscript. L279-L301, L326-327, L408-L416, L426-439.
The authors must investigate the quality of the films and their reproducibility.
Response: The quality of the film was investigated by studying the mechanical property of coated charred documents using starch analog and non-coated charred documents as per the reviewer’s comment. Moreover, the coating was done thrice on the same type of paper and pen combination for reproducible results of stabilization, preservation, and decipherment of charred documents.
Reviewer 4 Report
I thank the authors for their answers and this new manuscript version.
Could the authors clarify what they mean by "longpass" in the revised text? maybe they want to say nm?
I consider the authors' work well performed but the interest of the scientific community is quite limited since it adds nothing to the actual knowledge.
I cannot suggest its publication and I let the editor decide if it fulfil the journal standard.
English is understandable and mostly clear
Author Response
Reviewer 4
I thank the authors for their answers and this new manuscript version.
Could the authors clarify what they mean by "longpass" in the revised text? maybe they want to say nm?
Response: Longpass is the wavelength of light in nanometres (nm).
I consider the authors' work well performed but the interest of the scientific community is quite limited since it adds nothing to the actual knowledge.
Response: Sir, I appreciate your feedback, and understand your perspective on the interest of the scientific community.
However, I believe there are several positive aspects to consider about this research:
First and foremost, the author’s efforts in exploring new preservation techniques demonstrate their commitment to the field of forensic document decipherment and historical document preservation. The value of crucial documents examination, their restoration, and preservation is a very important aspect for questioned document examiners (QDE), as they encounter huge criminal cases related to damaged documents with fire, water, etc. QDE’s are asked to stabilize those fragile burnt or charred documents and to reveal the invisible writing on them in order to either know what was originally written on the document, match the questioned writings with those of suspected samples, and finally; who wrote them. This is done because, criminals usually burn the crucial documents linked to any fraud and think that once the documents get burnt, they cannot be recovered and the contents on them are lost forever. But the forensic mind they don’t know. Besides the fire, there are several other factors, that come into play while the documents are getting burnt, which prevent the documents to be burnt completely and turning into ash. Hence, those documents are examined by questioned documents examiners to recover the charred invisible content, which is related to any serious offense and may help to provide vital support in the case.
The conventionally used chemical in forensic science laboratories for charred document stabilization and preservation is Polyvinyl Acetate (PVA), but as stated in the manuscript, because of many demerits of PVA, for e.g., chemical nature, emission of toxic fumes, non-biodegradable, non-environment friendly, slow setting speed etc., an alternative to a this, was needed, often naturally prepared solution. Therefore, another significant merit of the authors' work lies in its potential practical application. Starch-based analog present a novel and eco-friendly approach to document preservation, which could have far-reaching implications for not only stabilizing, and preserving valuable documents of forensic significance but also providing appreciable decipherment soon after coating with the synthesized starch-based analog, which is not possible in case of PVA. PVA only aid in the stabilization of charred documents, moreover according to one research, PVA hinders decipherment under UV and IR light, which is not in the case of Starch-based analog. Moreover, the analog can also preserve historical records, artifacts, and cultural heritage for conservation significance. It's essential to continue supporting research that explores alternative preservation methods, as it contributes to the advancement of the field and opens up new avenues for safeguarding our past.
Furthermore, even if the current interest may be limited, it's worth noting that scientific breakthroughs have a cumulative effect. Each piece of research contributes to the overall knowledge base and can serve as a foundation for future studies and improvements. The authors' work might inspire other researchers to explore related avenues or refine and expand upon their findings, eventually generating more interest and recognition in the scientific community.
While it may be true that the scientific community's interest may seem limited at the moment, it's essential to remember that groundbreaking research often takes time to gain widespread recognition and appreciation.